# Aging Combined with High Waist-to-Hip Ratio Is Associated with a Higher Risk of Gastro-Esophageal Reflux Disease

**DOI:** 10.3390/jcm11175224

**Published:** 2022-09-04

**Authors:** Lo-Yip Yu, Ying-Chun Lin, Yang-Che Kuo, Hung-Ju Ko, Ming-Jen Chen, Horng-Yuan Wang, Shou-Chuan Shih, Chuan-Chuan Liu, Kuang-Chun Hu

**Affiliations:** 1Division of Gastroenterology, Department of Internal Medicine, Mackay Memorial Hospital, Taipei 10449, Taiwan; 2Healthy Evaluation Center, MacKay Memorial Hospital, Taipei 10449, Taiwan; 3Department of Anesthesiology, MacKay Memorial Hospital, Taipei 10449, Taiwan; 4MacKay Junior College of Medicine, Nursing, and Management, Taipei 10449, Taiwan

**Keywords:** elderly, waist-to-hip ratio (WHR), gastro-esophageal reflux disease (GERD)

## Abstract

**Background and Objective**: To assess whether the combination of high waist-to-hip ratio (WHR) and elderly age is associated with higher risk of GERD. **Material and Methods**: A total of 16,996 subjects aged ≥20 years who received esophagogastroduodenoscopy (EGD) between January 2010 and December 2019. We evaluated the risk of GERD in different age groups and WHR groups in unadjusted analysis and multivariate logistic regression models for predictors of GERD. **Results:** There was a trend towards more participants with both age ≥65 years and WHR ≥ 1 (*n* = 129) (*n* = 66, 51%) than participants with age < 65 and WHR < 0.9 (*n* = 10,422) (*n* = 2814, 27%) presenting with GERD. Participants who had both age ≥ 65 years and high WHR ≥ 1 had the highest risk of any type of GERD (adjusted OR, 2.07; 95% CI, 1.44–2.96, *p* value < 0.05) based on multivariate logistic regression analysis. **Conclusions:** The combination of having a high WHR and being elderly was associated with a higher risk of GERD, and preventing central obesity in the elderly population reduced the risk of GERD and the requirement for medical resources.

## 1. Introduction

Gastro-esophageal reflux disease (GERD) is a condition that develops when the reflux of stomach content causes either symptoms or an array of potential esophageal and extra-esophageal manifestations. A prevalence of 10~20% in the western world making GERD is highly prevalent worldwide [1]. At some time during the course of a year, 60% of persons suffered from GERD and 20~30% of persons suffered at least annually [2]. Typical symptoms of GERD are regurgitation and heartburn. Extra-esophageal syndromes include chronic cough, laryngitis, and asthma [3]. An increasing incidence of GERD and its complications, including Barrett’s esophagus and adenocarcinoma of the esophagus, throughout the world has been found [4].

GERD has a direct impact on quality of life, especially in the elderly [5]. In elderly individuals, Zhu et al. observed that abnormal gastroesophageal reflux occurred more frequently than in younger individuals. The mean prevalence of pH < 4 was significantly higher (32.5%) than that found in younger patients (12.9%) [6]. Age-related progressive impairment of esophageal clearance and degradation of the gastroesophageal junction results in GERD [7]. From the age of 40 years onward, Gregersen et al. found that esophageal function naturally deteriorates [8]. A total of 60% of patients older than 60 years of age have hiatal hernia and the presence of hiatal hernia has been associated with lower esophageal sphincter dysfunction [9].

Body mass index (BMI) has been used as an indicator of obesity. Obesity has been known to promote GERD. There is strong evidence from epidemiological studies of populations with GERD symptoms showing associations with high prevalence rates of obesity [10]. However, the pattern of obesity may be more important than BMI alone [11]. Studies have shown that central obesity relates more to reflux symptoms than peripheral deposition of fat among patients in obesity [12]. Edelstein et al. reported an association between a high waist-to-hip ratio (WHR) and GERD [13]. Ringhofer et al. reported that WHR was a better predictor of esophageal acid exposure than BMI [14].

In this retrospective cross-sectional study, we enrolled a large sample of apparently healthy people who underwent routine health checkups. We hypothesized that the combination of high WHR and aging was associated with a higher riskof GERD.

## 2. Material and Methods

### 2.1. Patient Selection

We retrospectively analyzed the data of 23,257 subjects who had underwent annual routine checkups at MacKay Memorial Hospital, Taipei, Taiwan, between January 2010 and December 2019. The inclusion criteria were as follows: individuals aged ≥ 20 years and individuals who had undergone esophagogastroduodenoscopy (EGD) on the same day as part of a health checkup were enrolled for analysis.

The exclusion criteria were as follows:

Previous gastro-esophageal surgery.Stomach or esophagus with previous or current cancer.Presence of gastric outlet obstruction.Use of medications (including nitrates, anti-cholinergic agents,

Benzodiazepines, and calcium channel blockers) interfering with GERD.

Current use of proton pump inhibitors or histamine-2 receptor antagonists.Known specific esophageal motor disorder.Incomplete data.

A total of 16,996 study participants (8951 males and 8045 females) were enrolled for further study.

### 2.2. Clinical Data Collection and Questionnaire

Baseline characteristics, including age, sex, height, weight, body mass index (BMI), body fat, waist circumference (WC), hip circumference (HC), WHR, personal medical history, and current medicine use, were obtained from a questionnaire completed at the time of the physical check-up. BMI was calculated as the ratio between weight (kg) and the square of height (m^2^). WC was measured in centimeters, at a midpoint between the lowermost rib and the anterior superior iliac spine with a non-elastic tape. HC was measured in centimeters at the widest portion of the hip. WHR was calculated as the ratio of WC divided by HC. Blood samples were collected from all subjects after 12 h of fasting, and the parameters measured included a complete blood count (CBC), fasting plasma glucose AC, hemoglobin A1C (HbA1c), albumin, total cholesterol, triglycerides (TGs), low-density lipoprotein (LDL), high-density lipoprotein (HDL), glutamic oxaloacetic transaminase (GOT), blood urea nitrogen (BUN), creatinine, and uric acid, and the blood samples were obtained from the participants on the same health checkup day as when the EGDs were performed.

### 2.3. Scanning Protocol and GERD Severity Evaluation

All EGDs were performed at the Health Evaluation Center. Some people had gastrointestinal discomfort, but most of the others did not show any discomfort clinically. The definition of having GERD was determined by EGDs findings, rather than being defined by the symptoms of clinically presenting GERD. Four experienced gastroenterologists blinded to the study aimed to perform the EGD using gastrofiberscopes (GIFQ260, Olympus Optical, Tokyo, Japan). The severity of GERD was graded from A to D according to the Los Angeles Classification.

Grade A: One (or more) mucosal break, no longer than 5 mm that does not extend between the tops of two mucosal folds.

Grade B: One (or more) mucosal break, more than 5 mm long that does not extend between the tops of two mucosal folds.

Grade C: One (or more) mucosal break that is continuous between the tops of two or more mucosal folds but involves less than 75% of the circumference.

Grade D: One (or more) mucosal break involving at least 75% of the esophageal circumference.

### 2.4. Statistical Analysis

Demographic variables were recorded for each subject. When the data for continuous variables fit a standard Gaussian distribution, a *t*-test was applied to compare GERD and non-GERD participants. The data were expressed as the mean ± standard deviation (SD). When the data for continuous variables failed to fit a normal distribution, a rank-sum test was applied to compare the two groups of participants. The data were expressed as the median ± interquartile range (IQR). Categorical variables are expressed as numbers (percentages), and the chi-squared test was adopted. Univariable and multivariable logistic regression analyses were applied to calculate the odds ratios (ORs) with 95% confidence intervals (CIs). Unadjusted ORs and ORs adjusted for clinically relevant predictors of GERD were presented. All variables with *p* < 0.05 were considered statistically significant. All analyses were performed using R version 4.1.2 (R Core Team, Vienna, Austria, 2021).

## 3. Results

### 3.1. Patient Characteristics

Table 1 shows the demographic data of patients with and without GERD. There were significant differences in all variables except body fat and platelet count.

### 3.2. Risk of Any Type of GERD According to Different Age Groups

Table 2 shows the prevalence of GERD in different age groups. A total of 2261 of 7567 participants (30%) aged < 45 years presented with GERD. For age 45 to <55, 1439 of 4358 participants (33%) presented with GERD (adjusted OR, 1.13; 95% CI, 1.04–1.23, *p* value < 0.05). For age 55 to <65, 1288 of 3834 participants (34%) presented with GERD (adjusted OR, 1.15; 95% CI, 1.05–1.25, *p* value < 0.05). A total of 288 of 765 participants (38%) aged 65 to <70 years presented with GERD (adjusted OR, 1.35; 95% CI, 1.16–1.58, *p* value < 0.05), and 180 of 466 participants (39%) aged ≥ 70 years presented with GERD (adjusted OR, 1.41; 95% CI, 1.16–1.72, *p* value < 0.05). We observed a trend that a higher proportion of participants with older age presented with GERD.

### 3.3. Risk of Any Type of GERD according to Different WHR Groups

Table 3 shows the prevalence of GERD in the different WHR groups. A total of 2968 of 10,769 participants (28%) with WHR < 0.9 presented with GERD. For individuals with WHR 0.9 to 0.99, 2222 of 5605 participants (40%) presented with GERD (adjusted OR, 1.32; 95% CI, 1.21–1.43, *p* value < 0.05). For individuals with a WHR ≥ 1, a total of 262 of 590 participants (44%) presented with GERD (adjusted OR, 1.45; 95% CI, 1.20–1.74, *p* value < 0.05). We observed a trend toward a higher proportion of participants with high WHR presenting with GERD.

### 3.4. Risk of GERD According to Different Age Groupsand WHR Groups

Table 4 shows that of the total subset of participants with age < 65 and WHR < 0.9 (*n* = 10,422), 2814 (27%) presented with GERD. A total of 154 of 453 participants (34%) with an age ≥ 65 years and WHR < 0.9 presented with GERD, and 1975 of 4938 participants (40%) with an age < 65 years combined with WHR 0.9 to 0.99 presented with GERD. Regarding individuals with an age ≥ 65 years and WHR 0.9 to 0.99, 247 of 633 participants (39%) presented with GERD, while 196 of 456 participants (43%) with an age < 65 years and WHR ≥ 1 presented with GERD. However, we observed a trend that a higher proportion of participants with both age ≥ 65 years and WHR ≥ 1 presented with GERD (66 of 129 participants, 51%).

### 3.5. Multivariate Logistic Regression for Predictors of GERD

Table 4 shows the impact of aging and WHR on the risk of GERD. We classified participants based on WHR and old age (<65 or ≥65). The risk of any GERD was higher in participants who were an older age or higher WHR than in those without either condition. Participants with an age ≥ 65 years and WHR ≥ 1 had the highest risk of GERD (adjusted OR, 2.07; 95% CI, 1.44–2.96, *p* value < 0.05) in the multivariate logistic regression. Therefore, the combination of old age and higher WHR is associated with a higher risk of GERD.

## 4. Discussion

One of the most prevalent clinical conditions affecting the gastrointestinal tract is GERD [15]. Severe symptoms frequent of GERD are associated with impaired health-related quality of life and time lost from work, further emphasizing the clinical significance of this entity [16]. The associations between GERD symptoms and age, BMI, alcohol consumption, smoking, stress, anti-cholinergic drugs, aspirin, and non-steroidal anti-inflammatory drugs have been identified by previous population-based studies [17]. To the best of our knowledge, this report is the first to indicate a positive correlation between aging and high WHR complicated with GERD. Such a positive correlation was noticed when individuals were older and had a higher WHR. In multivariate logistic regression of predictors of GERD, participants with an age ≥ 65 years and WHR ≥1 had the highest risk of GERD (adjusted OR, 2.07; 95% CI, 1.44–2.96, *p* value < 0.05). One strength of our study is that it included a large sample size (16,996 subjects).

In elderly patients, GERD is the most common upper gastrointestinal disorder [18] and up to 40% of the adult United States population are affected at least once a month [19]. In Finland, a population-based study revealed that the prevalence of daily GERD symptoms in those aged 65 years and older was 8% in men and 15% in women [20]. In the elderly population, the presentation of GERD is not “classic” and the severity of reported symptoms does not correlate with the severity of mucosal disease [3]. Some studies have found that up to 50% of cases of non-cardiac chest pain may have an association with GERD [21]. Extra-esophageal complications of GERD occur more commonly in elderly individuals. These complications include laryngitis, asthma, atypical chest pain that can simulate angina pectoris, chronic cough, and pulmonary aspiration [3]. Despite a trend toward increased acid exposure complicated with esophageal mucosal injury, elderly individuals tend to have less symptom perception [22]. Elderly patients are at greater risk than younger patients for developing serious complications of GERD, such as Barrett’s esophagus, esophageal stricture, and esophageal adenocarcinoma [23]. More elderly patients (20.8%) have grade C/D of GERD compared with only 3.4% of younger patients [6].

In our study, we observed a trend toward a higher proportion of participants with older age presenting with GERD. The highest proportion of patients suffering from GERD (180 of 466 participants, 39%) was found among individuals with an age ≥ 70 years (adjusted OR, 1.41; 95% CI, 1.16–1.72, *p* value < 0.05) compared with the lowest proportion (2261 of 7567 participants, 30%) found in individuals aged < 4 5 years. The development of GERD is led by multiple factors. First, from the age of 40 years onwards, esophageal function naturally deteriorates, which may increase the risk for the development of GERD [8]. In elderly patients, compared to the situation in younger individuals, there is a significant decrease in the amplitude of peristaltic contraction and an increase in the frequency of non-propulsive contractions, causing esophageal motility disturbances [24]. Second, an effective buffer that helps to re-establish a more alkaline pH in the esophageal lumen is saliva [25]. Decreasing salivary production with age, which is associated with a significantly decreased salivary bicarbonate response to acid perfusion within the esophagus, causing impairment of esophageal acid clearance [26]. Third, in elderly patients, delayed gastric emptying and duodenogastric reflux of bile plays a significant role in GERD pathogenesis [27]. With increasing age, most data support the finding that there is a decrease in gastric emptying time [28]. Fourth, when patients are in their fifth and sixth decade, reduced pain perception begins causing the ability to sense acid in the esophagus to fade. This is a progressive phenomenon that becomes more profound as patients reach their seventh and eighth decade. A possible explanation for the greater number of elderly patients who have reflux without symptoms is an age-related loss of neurons in the myenteric plexus of the esophagus [22]. Fifth, a hiatal hernia involves the displacement of the lower esophageal sphincter toward the thoracic cavity and impairs the clearance of refluxate from the esophagus [29]. In 60% of patients older than 60 years, hiatal hernia has been identified [9]. In another study, the presence of hiatal hernia and the size of the hiatus increased with age [30].

Other factors that lead to the development of GERD in elderly individuals includecomorbidities and medications, which can diminish esophageal sphincter tone and esophageal clearance mechanisms [31]. In elderly individuals, multiple medications including calcium channel blockers, benzodiazepines, nitrates, and antidepressants are more frequently taken for comorbid illnesses, such as hypertension, cardiovascular disease, and depression, which are well known to decrease lower esophageal sphincter (LES) pressure [32]. Other medications, including potassium tablets, iron supplements, and non-steroidal anti-inflammatory drugs, have direct effects on esophageal injury, and are more frequently used by elderly individuals [5]. Moreover, elderly individuals undergo lifestyle changes that exacerbate reflux, including reduced mobility, the development of obesity, an increased sedentary lifestyle, and increased recumbency due to comorbid illness, such as cerebrovascular disease, dementia, and pulmonary disease [28].

In our study, we observed a trend towards a higher proportion of participants with high WHR presenting with GERD. The highest proportion of individuals suffering from GERD (262 of 590 participants, 44%) was found among participants with WHR ≥ 1 (adjusted OR, 1.45; 95% CI, 1.20–1.74, *p* value < 0.05) compared with the lowest (2968 of 10,769 participants, 28%) among participants with WHR < 0.9. A risk factor for GERD symptoms, esophageal erosions and esophageal adenocarcinoma, is obesity as measured by BMI [33]. However, BMI is not always an accurate estimate of adiposity, particularly in men with a greater muscle mass. For the same BMI, it is known that the distribution of body fat tends to be more visceral than truncal in groups with a high risk of GERD [11]. WHR showed a stronger association with esophageal acid exposure than BMI. Of endoluminal pH < 4 in the distal esophagus, Ringhof et al., showed that 6.9% of cases are attributable to WHR. Furthermore, WHR showed an association with impaired esophageal acid clearance was observed [14]. Lee et al., showed a significant increase in the risk of GERD with obesity, specifically abdominal obesity measured by the WHR. Their study suggested that the development of GERD may be the pattern of obesity more important than BMI [11].

The association of central obesity with GERD could be explain by several hypotheses. The LES acts as a physiological barrier and prevent refluxing gastric contents to the esophagus [34]. The contribution of a higher WC to GERD involves a mechanical pathogenesis by abdominal adipose tissue causing extrinsic gastric compression, which would tend to increase intra-gastric pressure, thereby leading to acid reflux by inducing relaxation of the lower esophageal sphincter [35]. Hiatal hernia formation and enlargement is another possible explanation for GERD. A total of 1389 patients undergoing upper endoscopy in a retrospective analysis, obesity was a significant independent risk factor for hiatal hernia [36]. A strong association between waist circumference and increased separation of gastroesophageal pressure components (indicative of enlargement of hiatal hernia) have been shown by previous studies with GERD patients or healthy individuals [37]. Among people with obesity, abdominal fat is implicated as the cause of GERD. Visceral abdominal fat has been closely associated with elevated serum levels of interleukin-6, leptin, and tumor necrosis factor-X. These cytokines are proinflammatory cytokines and enhance gastric secretory function and may influence the integration of gastroesophageal junctions, and they have been shown in multiple studies to be overexpressed in GERD patients [38].

Regarding elderly individuals, it is unclear whether it is better to have higher or lower adiposity. In the past, there were still some controversies in the medical field. Poor survival with a higher prevalence of coronary heart disease and type 2 diabetes mellitus in the elderly population is associated with obesity after accounting for confounders [39]. However, a drop in all-cause mortality in survivors of sudden cardiac arrest correlates with higher BMI, suggesting that the obesity paradox also prevails in the post-arrest condition [40]. In our study, we found that aging combined with high WHR will increase the risk of GERD and affect the quality of life of elderly individuals. Aging is an unavoidable event, so avoiding central obesity is very important in the care of elderly individuals.

Our study has several important limitations. First, this was a retrospective observational study that included only participants with relatively high incomes and higher health awareness, which may not represent the general population. Second, this was a single medical center study, which may have resulted in selection bias. Third, the severity of GERD was graded from A to D according to the Los Angeles Classification. Our study reported only the incidence of GERD in relation to WHR and age but did not take into consideration the severity of GERD, which is a study limitation. Despite these limitations, because of the large number of participants in this study, these biases have been statistically reduced as much as possible.

## 5. Conclusions

High WHR and aging alone can cause GERD. In our study, a high WHR combined with aging is associated with a higher risk of GERD. Given the high prevalence of high WHR in developed countries and the growing elderly population, we would like to emphasize the importance of preventing central obesity in the elderly population and reducing the risk of GERD and the use of medical resources.

## Figures and Tables

**Table 1 jcm-11-05224-t001:** Baseline characteristics (presented with mean and SD, median and IQR, or *n* and %).

Variables	Without GERD (*n* = 11,537)		With GERD (*n* = 5459)	*p*-Value	
**Age, mean (SD), year**	47.00	(39, 56)	49.00	(40, 57)	0.000
Sex (Male %)	5563	48%	3388	62%	0.000
Height, mean (SD), cm	164.354	8.598	166.034	8.707	<0.001
Weight, mean (SD), kg	63.602	12.728	67.250	13.494	<0.001
BMI, mean (SD), kg/m^2^	23.418	3.572	24.257	3.727	<0.001
Body fat, mean (SD), %	25.485	6.720	25.617	6.537	0.228
Waist, mean (SD), cm	81.403	9.994	84.467	10.268	<0.001
Buttock, mean (SD), cm	94.454	6.465	95.529	6.788	<0.001
Waist-to-hip ratio (WHR), mean (SD)	0.860	0.071	0.883	0.071	<0.001
HbA_1c_, mean (SD), %	5.50	(5.2, 5.7)	5.50	(5.3, 5.8)	0.000
Glucose AC, mean (SD), mg/dL	94.00	(88, 99)	95.00	(89, 102)	0.000
Albumin, mean (SD), g/dL	4.575	0.272	4.610	0.271	0.000
Total cholesterol, mean (SD), mg/dL	199.757	37.472	202.696	39.616	0.000
Triglyceride, mean (SD), mg/dL	95.00	(68, 141)	109.00	(75, 161)	0.000
LDL, mean (SD), mg/dL	128.032	34.794	130.106	35.191	0.000
HDL, mean (SD), mg/dL	55.059	16.010	51.946	15.266	<0.001
GOT, mean (SD), IU/L	22.00	(19, 27)	23.00	(19, 28)	0.000
BUN, mean (SD), mg/dL	9.00	(7, 11)	9.00	(7, 11)	0.000
Creatinine, mean (SD), mg/dL	0.80	(0.7, 1)	0.90	(0.7, 1)	0.000
Uric acid, mean (SD), mg/dL	5.653	1.415	5.989	1.448	<0.001
WBC count, mean (SD), ×10^3^/uL	6.078	1.667	6.278	1.815	0.000
Hb, mean (SD), g/dL	14.001	1.592	14.363	1.576	<0.001
Ht, mean (SD), %	41.938	4.486	42.980	4.456	<0.001
MCV, mean (SD), fl	88.415	7.193	88.798	6.921	0.001
Platelet count, mean (SD), ×10^3^/uL	255.070	60.689	256.182	60.593	0.264

BMI: body mass index; HbA_1c_: glycated hemoglobin; LDL: Low-density lipoprotein; HDL: High-density lipoprotein; GOT: Glutamic Oxaloacetic Transaminase; BUN: Blood urea nitrogen; WBC: White blood cell; Hb: Hemoglobin; Ht: Hematocrit; MCV: Mean corpuscular volume.

**Table 2 jcm-11-05224-t002:** Different ages and their risk of GERD.

	Number	GERD (+)	GERD (%)	OR (Unadjusted)		95% CI	*p*-Value	OR (Adjusted)	95% CI		*p*-Value
Age <45	7567	2261	30	reference	-	-	-	reference		-	-
Age 45 to <55	4358	1439	33	1.157	1.068	1.253	0.000	1.131	1.043	1.228	0.003
Age 55 to <65	3834	1288	34	1.187	1.092	1.290	0.000	1.145	1.051	1.248	0.002
Age 65 to <70	765	288	38	1.417	1.214	1.653	0.000	1.352	1.155	1.583	0.000
Age ≥ 70	466	180	39	1.477	1.218	1.791	0.000	1.408	1.156	1.715	0.001

Adjusted for AC Glucose, Sex, and BMI.

**Table 3 jcm-11-05224-t003:** Different WHR with the risk of GERD.

	Number	GERD (+)	GERD (%)	OR (Unadjusted)	95% CI		*p*-Value	OR (Adjusted)	95% CI		*p*-Value
WHR < 0.9	10,769	2968	28	reference				reference			
WHR0.9 to 0.99	5605	2222	40	1.726348	1.613	1.848	<2 × 10^−16^	1.318	1.213	1.432	0.000
WHR ≥ 1	590	262	44	2.09949	1.775	2.483	<2 × 10^−16^	1.447	1.200	1.744	0.000

Adjusted for AC Glucose, Sex, and BMI.

**Table 4 jcm-11-05224-t004:** Interaction of WHR and age with the risk of GERD.

		Any GERD									
		N	%	Unadjusted OR	95% CI		*p*-value	Adjusted OR	95% CI		*p*-Value
WHR < 0.9	Age < 65	2814	27	reference				reference			
WHR < 0.9	Age ≥ 65	154	34	3.838	5.157	1.003	0.003	1.394	1.141	1.703	0.001
WHR0.9 to 0.99	Age < 65	1975	40	5.803	6.610	1.000	0.000	1.329	1.219	1.449	0.000
WHR0.9 to 0.99	Age ≥ 65	247	39	5.380	7.278	1.000	0.000	1.365	1.150	1.620	0.000
WHR ≥ 1	Age < 65	196	43	7.172	10.823	1.000	0.000	1.329	1.079	1.638	0.007
WHR ≥ 1	Age ≥ 65	66	51	16.292	52.039	1.000	0.000	2.068	1.444	2.962	0.000

Adjusted for AC glucose, sex, and BMI.

## Data Availability

Not applicable.

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
