# Peer review of "Aging Combined with High Waist-to-Hip Ratio Is Associated with a Higher Risk of Gastro-Esophageal Reflux Disease"

_jcm, 2022, doi:10.3390/jcm11175224_

Round 1

Reviewer 1 Report

The authors describe an interesting study on aging and waist to hip ratio in relation to GERD and find that an older age combined with a higher ratio is associated with a higher risk of GERD. The strength of this study is definitely the large sample size.

Comments:

1. Why were the patients getting EGDs? Did they have any symptoms?

2. You mentioned how the severity of GERD was graded but do not comment on how having "GERD" was defined. Please include this in the methods.

3. You mention that you adjusted for clinically relevant predictors of GERD but do not tell us what you adjusted for. What variables were included in the multivariate analysis? 

4. In your results, you describe the results saying patients "suffered from GERD" and you conclude each paragraph by saying "patients presented with GERD". I think you should be consistent and use the same type of language throughout. 

5. Since this is a cross-sectional study, causation cannot be determined and you should comment only on associations. I would suggest you delete line 17 on page 10/15 (last sentence in the results under multivariate logistic regression) "Therefore the combination of old age and higher WHR increases the risk of GERD". In general, to describe this correlation, I would recommend you use "was associated with a higher risk of GERD". 

6. In line with the above comment, I would recommend you modify the conclusion and change "increased the risk of GERD" to "was associated with a higher risk of GERD". 

7. In the abstract, the first sentence of the conclusion has a typo and should be modified to read "having a high WHR and being elderly...". I would also suggest the whole thing gets reworded to reflect association rather than causation "The combination of having a high WHR and being elderly was associated with a higher risk of GERD. 

8. Introduction line 34 (Harrison) - I think you missed adding the reference.

Author Response

To the Editor: 

    We appreciate the opportunity to address the Reviewer’ comments and revise our manuscript “jcm-1877844 - Aging combined with high waist-to-hip ratio is associated with a higher risk of gastroesophageal reflux disease”. We had followed reviewers’ suggestion and modified our manuscript accordingly, as shown in highlighting.

Thanks again for your kindly review of our article.

Sincerely,

Kuang-Chun Hu, M.D
Division of Gastroenterology, Department of Internal Medicine, Mackay Memorial Hospital, Taipei 10449, Taiwan

E-mail address: mimiandbear2001@yahoo.com.tw

Response to Reviewer’s comments

Manuscript ID: jcm-1877844

Reviewer’s Comments

Our Response

Location of edits

Reviewer 1:

Why were the patients getting EGDs? Did they have any symptoms?

We appreciate the reviewer’s useful question. All EGDs were performed at the Health Evaluation Center. The source of customers (cannot be said to be patients) is a group with relatively high health awareness and high economic ability. They undergone routine health checkups. Some people had gastrointestinal discomfort, but most of the others did not show any discomfort clinically.

The customers may not represent the general population making our study design limitation.

highlighting in article: section ”Material and Method- scanning protocol and GERD severity evaluation “and “Discussion” about limitation”

You mentioned how the severity of GERD was graded but do not comment on how having "GERD" was defined. Please include this in the methods.

We thank the reviewer for this helpful suggestion. The definition of having GERD was determined by EGDs findings, rather than being defined by the symptoms of clinically presenting GERD. We include this in the methods.

highlighting in article: section ”Material and Method- scanning protocol and GERD severity evaluation”

You mention that you adjusted for clinically relevant predictors of GERD but do not tell us what you adjusted for. What variables were included in the multivariate analysis?

We appreciate the important point about for what variables were included in the multivariate analysis.  The variables for adjusted including AC glucose, sex, and BMI.

highlighting in Table 2 to Table 4

In your results, you describe the results saying patients "suffered from GERD" and you conclude each paragraph by saying "patients presented with GERD". I think you should be consistent and use the same type of language throughout.

We thank and agree with the reviewer suggestion that consistent and use the same type of language throughout is important. We rewrite all the sentences from "suffered from GERD" to "patients presented with GERD"

Highlighting all the changes in article: section “Results”

Since this is a cross-sectional study, causation cannot be determined and you should comment only on associations. I would suggest you delete line 17 on page 10/15 (last sentence in the results under multivariate logistic regression) "Therefore the combination of old age and higher WHR increases the risk of GERD". In general, to describe this correlation, I would recommend you use "was associated with a higher risk of GERD".

We appreciate the reviewer’s very useful opinion We revise our manuscript title and all the sentences and use "is associated with a higher risk of GERD".

Highlighting all the changes in article

In line with the above comment, I would recommend you modify the conclusion and change "increased the risk of GERD" to "was associated with a higher risk of GERD".

We appreciate the reviewer’s very useful opinion We revise our manuscript title and all the sentences and use "is associated with a higher risk of GERD".

Highlighting all the changes in article

In the abstract, the first sentence of the conclusion has a typo and should be modified to read "having a high WHR and being elderly...". I would also suggest the whole thing gets reworded to reflect association rather than causation "The combination of having a high WHR and being elderly was associated with a higher risk of GERD.

We appreciate the reviewer’s very useful opinion We revise the abstract conclusion as “The combination of having a high WHR and being elderly is associated with a higher risk of GERD,

Highlighting in the abstract conclusion

Introduction line 34 (Harrison) - I think you missed adding the reference.

We thank the reviewer reminding. We add the reference as follow: Extraesophageal syndromes include chronic cough, laryngitis and asthma (3).

Highlighting the changes in section “Introduction”

Reviewer 2 Report

Thank you for allowing me to review this paper. The authors report the impact of age and waist-to-hip ratio on the prevalence of gastroesophageal reflux disease.  The authors included 16996 subjects from 2010 to 2019. The authors state that the gastroenterologist who performed the GI endoscopy was not aware of the diagnosis. Was he alone and did he perform all examinations from 2010 to 2019?

The severity of GERD was graded from A to D according to the Los Angeles classification. Why was this classification not used by the authors to specify risk factors for severe GERD?

Could the authors provide data on smoking and alcohol consumption in each group with and without GERD?

Waist to hip ratio may vary by sex and in women by hormonal impregnation. Did the authors assess the impact of menopause in women in this study?

One of the main limitations of this study is the absence of gastroesophageal reflux symptoms. Could the authors shed some light on gastroesophageal reflux disease symptomatology in the patients included in this study?

Author Response

To the Editor: 

    We appreciate the opportunity to address the Reviewer’ comments and revise our manuscript “jcm-1877844 - Aging combined with high waist-to-hip ratio is associated with a higher risk of gastroesophageal reflux disease”. We had followed reviewers’ suggestion and modified our manuscript accordingly, as shown in highlighting.

Thanks again for your kindly review of our article.

Sincerely,

Kuang-Chun Hu, M.D
Division of Gastroenterology, Department of Internal Medicine, Mackay Memorial Hospital, Taipei 10449, Taiwan

E-mail address: mimiandbear2001@yahoo.com.tw

Response to Reviewer’s comments

Manuscript ID: jcm-1877844

Reviewer’s Comments

Our Response

Location of edits

Reviewer :

The authors state that the gastroenterologist who performed the GI endoscopy was not aware of the diagnosis. Was he alone and did he perform all examinations from 2010 to 2019?

We thank the reviewer for this helpful question.

“An experienced gastroenterologist blinded to the study aims performed the EGD” caused the reviewer's misunderstanding.

è Rewrite: Four experienced gastroenterologist blinded to the study aims performed the EGD

highlighting in article: section ”Material and Method- scanning protocol and GERD severity evaluation “

The severity of GERD was graded from A to D according to the Los Angeles classification. Why was this classification not used by the authors to specify risk factors for severe GERD?

We appreciate the reviewer’s very useful opinion. Severe GERD can refer to the findings under the GI endoscopy, or patient clinical presentation. We add the definition of GERD in the section: Material and method. In our study, the definition of having GERD was determined objective findings by GI endoscopy, rather than being defined by the symptoms of clinically presenting GERD.

Los Angeles classification was not used to specify risk factors for severe GERD is our study limitation. In “Discussion” about limitation:

è the severity of GERD was graded from A to D according to the Los Angeles Classification. Our study reported only the incidence of GERD in relation to WHR and age but did not take into consideration the severity of GERD, which is a study limitation.

highlighting in article: section ”Material and Method- scanning protocol and GERD severity evaluation

and “Discussion” about limitation.

Could the authors provide data on smoking and alcohol consumption in each group with and without GERD?

We appreciate the reviewer’s useful suggestion. Although health questionnaire asked people whether they smoke or alcohol consumption, the details of each person's answers were different. Such as smoking, Smoking every day? How many cigarettes every day?  Smoking years for how long and whether they quit smoking in the middle. For alcohol consumption:  How often do you drink? How much do you drink each time? What kind of alcohol do you drink? The details of each person's answer are very different. Therefore, we categorize smoking and alcohol consumption in the incomplete data in the exclusion criteria. Of course, it is also because our questionnaire design is not rigorous enough, which will cause the people answer varies.

highlighting in article: section ”Material and Method- patient selection“

Waist to hip ratio may vary by sex and in women by hormonal impregnation. Did the authors assess the impact of menopause in women in this study?

This is a good suggestion, but because our questionnaire does not ask about menopause, it cannot be assessed. In addition, gender has been included in the assessment in the multivariate logistic regression analysis.

Reviewer provides a good direction for future discussion about the impact of menopause in women associated with waist to hip ration and GERD presentation.

highlighting in Table 2 to Table 4

One of the main limitations of this study is the absence of gastroesophageal reflux symptoms. Could the authors shed some light on gastroesophageal reflux disease symptomatology in the patients included in this study?

We thank the reviewer useful opinion. The definition of having GERD in our study was determined by EGDs findings, rather than being defined by the symptoms of clinically presenting GERD

All GI endoscopy were performed at the Health Evaluation Center. The source of customers (can not be said to be patients) is a group with relatively high health awareness and high economic ability, which may not represent the general population. They undergone routine health checkups. Some people have gastrointestinal discomfort, but most of the others do not show any discomfort clinically.

The customers may not represent the general population making our study design limitation.

highlighting in article: section ”Material and Method- scanning protocol and GERD severity evaluation and “Discussion” about limitation.

Round 2

Reviewer 2 Report

the authors responded point by point to the comments and remarks.